# SketchGen: Generating Constrained CAD Sketches

**Wamiq Reyaz Para**[1]    **Shariq Farooq Bhat**[1]    **Paul Guerrero**[2]

**Tom Kelly**[3]    **Niloy Mitra**[2,4]    **Leonidas Guibas**[5]    **Peter Wonka**[1]

[1] KAUST    [2] Adobe Research    [3] University of Leeds
[4] University College London    [5] Stanford University

## Abstract

Computer-aided design (CAD) is the most widely used modeling approach for technical design. The typical starting point in these designs is 2D sketches which can later be extruded and combined to obtain complex three-dimensional assemblies. Such sketches are typically composed of parametric primitives, such as points, lines, and circular arcs, augmented with geometric constraints linking the primitives, such as coincidence, parallelism, or orthogonality. Sketches can be represented as graphs, with the primitives as nodes and the constraints as edges. Training a model to automatically generate CAD sketches can enable several novel workflows, but is challenging due to the complexity of the graphs and the heterogeneity of the primitives and constraints. In particular, each type of primitive and constraint may require a record of different size and parameter types. We propose SketchGen as a generative model based on a transformer architecture to address the heterogeneity problem by carefully designing a sequential language for the primitives and constraints that allows distinguishing between different primitive or constraint types and their parameters, while encouraging our model to re-use information across related parameters, encoding shared structure. A particular highlight of our work is the ability to produce primitives linked via constraints that enables the final output to be further regularized via a constraint solver. We evaluate our model by demonstrating constraint prediction for given sets of primitives and full sketch generation from scratch, showing that our approach significantly out performs the state-of-the-art in CAD sketch generation.

## 1   Introduction

Computer Aided Design (CAD) tools are popular for modeling in industrial design and engineering. The employed representations are popular due to their compactness, precision, simplicity, and the fact that they are directly 'understood' by many fabrication procedures.

CAD models are essentially a collection of primitives (e.g., lines, arcs) that are held together by a set of relations (e.g., collinear, parallel, tangent). Complex shapes are then formed by a sequence of CAD operations, forming a base shape in 2D that is subsequently lifted to 3D using extrusion operations. Thus, at the core, most CAD models are coplanar constrained sketches, i.e., a sequence of planar curves linked by constraints. For example, CAD models in some of the largest open-sourced datasets like Fusion360 [37] and SketchGraphs [27] are thus structured, or sketches drawn in algebraic systems like Cinderella [1]. Such sequences are not only useful for shape creation, but also facilitate easy editing and design exploration, as relations are preserved when individual elements are edited.

CAD representations are heterogeneous. First, different instructions have their unique sets of parameters, with different parameter counts, data types, and semantics. Second, constraint specifications involve links (i.e., pointers) to existing primitives or parts of primitives. Further, the same final shape

35th Conference on Neural Information Processing Systems (NeurIPS 2021).

can be equivalently realized with different sets of primitives and constraints. Thus, CAD models neither come in regular structures (e.g., image grids) nor can they be easily encoded using a fixed length encoding. The heterogeneous nature of CAD models makes it difficult to adopt existing deep learning frameworks to train generative models. One easy-to-code solution is to simply ignore the constraints and rasterize the primitives into an image. While such a representation is easy to train, the output is either in the image domain, or needs to be converted to primitives via some form of differential rendering. In either case, the constraints and the regularization parametric primitives provide is lost, and the conversion to primitives is likely to be unreliable. A slightly better option is to add padding to the primitive and constraint representations, and then treat them as fixed-length encodings for each primitive/constraint. However this only hides the difficulty, as the content of the representation is still heterogeneous, the generator now additionally has to learn the length of the padding that needs to be added, and the increased length is inefficient and becomes unworkable for complex CAD models.

We introduce SketchGen as a generative framework for learning a distribution of constrained sketches. Our main observation is that the key to working with a heterogeneous representation is a well-structured language. We develop a language for sketches that is endowed with a simple syntax, where each sketch is represented as a sequence of tokens. We explicitly use the syntax tree of a sequence as additional guidance for our generative model. Our architecture makes use of transformers to capture long range dependencies and outputs both primitives along with their constraints. We can aggressively quantize/approximate the continuous primitive parameters in our sequences as long as the inter-primitive constraints can be correctly generated. A sampled graph can then be converted, if desired, to a physical sketch by solving a constrained optimization using traditional solvers, removing errors that were introduced by the quantization.

We evaluate SketchGen on one of the largest publicly-available constrained sketch datasets Sketch-Graph [27]. We compare with a set of existing and contemporary works, and report noticeable improvement in the quality of the generated sketches. We also present ablation studies to evaluate the efficacy of our various design choices.

## 2 Related Work

**Vector graphics generation.** Vector graphics, unlike images, are presented in domain-specific languages (e.g., SVG) and are not in a format easily utilized by standard deep learning setups. Recently, various approaches have been proposed by linking images and vectors using differentiable renderer [19, 26] or supervised with vector sequences (e.g., deepSVG [4], SVG-VAE [20], and Cloud2Curve [5]). DeepSVG, mostly closely related to our work, proposes a hierarchical non-autoregressive model for generation, building on command, coordinate, and index embeddings.

**Structured model generation.** Geometric layouts are often represented as graphs encoding relationships between geometric entities. Motivated by the success in image synthesis, several authors attempted to build on generative adversarial networks [17, 22] or variational autoencoder [15]. Recently, autoregressive models like the transformer architecture [31] emerged as an important tool for generative modeling of layouts, e.g. [34, 23, 36]. Closely related to our work are DeepSVG [4] and Polygen [21]. These solutions are also related to modeling with variable length commands, but they simplify the representation so that commands are padded to a fixed length. Polygen is an auto-regressive generative model for 3D meshes that introduces several influential ideas. One of the proposed ideas, that we also build on, is to employ pointer networks [32] to refer to previously generated objects (i.e., linking vertices to form polygonal mesh faces). Unlike PolyGen, we work with heterogeneous primitives and constraints, and with edges that constrain the primitive parameters, motivating our constraint optimization step.

**CAD Datasets and CAD generation.** Recently, several notable datasets have emerged for CAD models: ABC [13], Fusion360 [37], and SketchGraphs [27]. Only the latter two include constraints and only SketchGraphs introduces a framework for generative modeling. There are several papers that focus on the boundary (surface) representation for CAD model generation without constraints such as UV-Net [9] and BRepNet [14]. A related topic is to recover (parse) a CAD-like representation from unstructured input data, e.g. [28, 6, 30, 18, 33, 35, 11, 3, 29, 40, 7, 10].

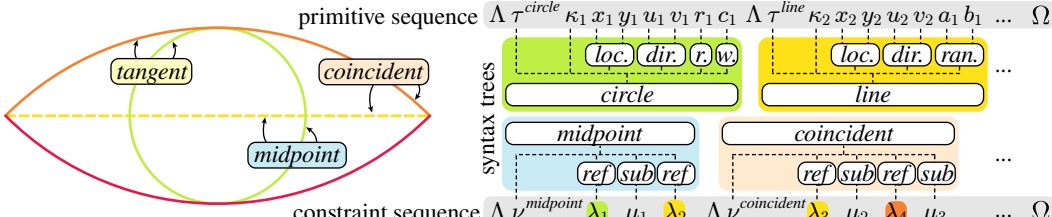

Figure 1: A typical CAD sketch consists of primitives such as lines, circles and arcs, and constraints between primitives, shown as floating boxes. We represent them as primitive- and constraint sequences in a language that is endowed with a simple syntax. We show the syntax trees of the first two primitives and constraints. Tokens in the constraint sequence reference primitives (colored token background).

**Concurrent work.** Multiple papers, namely Computer-Aided Design as Language [8], Engineering Sketch Generation for Computer-Aided Design [38], DeepCAD [39], have appeared on arXiv very recently, and should be considered to be concurrently works.

## 3 Overview

**CAD sketches.** A CAD sketch can be defined by a graph $S = (\mathcal{P}, \mathcal{C})$, where $\mathcal{P}$ is a set of parametric primitives, such as points, lines, and circular arcs and $\mathcal{C}$ is a set of constraints between the primitives, such as coincidence, parallelism, or orthogonality. See Figure 1 for an example. Some of the primitives are only used as construction aids and are not part of the final 3D CAD model (shown dashed in Figure 1). They can be used to construct more complex constraints; in Figure 1, for example, the center of the circle is aligned with the midpoint of the yellow line, which serves as a construction aid. Both primitives and constraints in the graph are heterogeneous: different primitives have different numbers and types of parameters and different constraints may reference a different number of primitives. Primitives $P \in \mathcal{P}$ are defined by a type and a tuple of parameters $P = (\tau, \kappa, \rho)$, where $\tau$ is the type of primitive, $\kappa$ is a binary variable which indicates if the primitive is a construction aid, and $\rho$ is a tuple of parameters with a length dependent on the primitive type. A point, for example, has parameters $\rho = (x, y)$. See Figure 2 for a complete list of primitives and their parameter type. Constraints $C \in \mathcal{C}$ are defined by a type and a tuple of target primitives $C = (\nu, \gamma)$, where $\nu$ is the constraint type and $\gamma$ is a tuple of references to primitives with a length dependent on the constraint type. Constraints can target either the entire primitive, or a specific part of the primitive, such as the center of a circle, or the start/end point of a line. In a coincidence constraint, for example, $\gamma = (\lambda_1, \mu_1, \lambda_2, \mu_2)$ refers to two primitives $P_{\lambda_1}$ and $P_{\lambda_2}$, while $\mu_1, \mu_2$ indicate which part of each primitive the constraint targets, or if it targets the entire primitive.

**CAD sketch generation with Transformers.** We show how a generative model based on the transformer architecture [31] can be used to generate CAD sketches defined by these graphs. Transformers have proven very successful as generative models in a wide range of domains, including geometric layouts [21, 36], but require converting data samples into sequences of tokens. The choice of sequence representation is a main factor influencing the performance of the transformer and has to be designed carefully. The main challenge in our setting is then to find a conversion of our heterogeneous graphs into a suitable sequence of tokens.

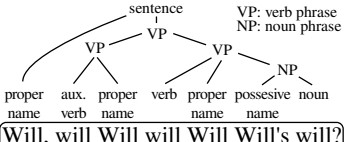

To this end, we design a language to describe CAD sketches that is endowed with a simple syntax. The syntax imposes constraints on the tokens that can be generated at any given part of the sequence and can help interpreting a sequence of tokens. An extreme example is the famous natural language sentence "*Will, will Will will Will Will's will?*", which is a valid sentence that is easier to interpret given its syntax tree, as shown in the inset. In natural language processing, syntax is complex and hard to infer automatically from a given sentence, so generative models usually only infer it implicitly in a data-driven approach. On the other end of the spectrum of syntactic complexity are geometric problems such as mesh generation [21], where the syntax consists of repeating triples of vertex coordinates or triangle indices that can easily be given explicitly, for example as indices from 1 to 3 that denote which element of the triple a given token represents. In our case, the grammar

is more complex due to the heterogeneous nature of our sketch graphs, but can still be stated explicitly. We show that giving the syntax of a sequence as additional input to the transformer helps sequence interpretation and increases the performance of our generative model for CAD sketches.

We describe our language for CAD sketches in Section 4. In this language, primitives are described first, followed by constraints. We train two separate generative models, one model for primitives that we describe in Section 5.1, and a second model for constraints that we describe in Section 5.2.

## 4 A Language for CAD Sketches

We define a formal language for CAD sketches, where each valid sentence is a sequence of tokens $Q = (q_1, q_2, \dots)$ that specifies a CAD sketch. A grammar for our language is defined in Figure 2, with production rules for primitives on the left and for constraints on the right. Terminal symbols for primitives include $\{\Lambda, \Omega, \tau, \kappa, x, y, u, v, a, b\}$ and for constraints $\{\Lambda, \nu, \lambda, \mu, \Omega\}$. Each terminal symbol denotes a variable that holds the numerical value of one token $q_i$ in the sequence $Q$. The symbols $\Lambda$ and $\Omega$ are special constants; $\Lambda$ marks the start of a new primitive or constraint, while $\Omega$ marks the end of the primitive or constraint sequence. $\tau$, $\nu$, $\kappa$, $\lambda$, and $\mu$ were defined in Section 3 and denote the primitive type, constraint type, construction indicator, primitive reference, and part reference type, respectively. The remaining terminal symbols denote specific parameters of primitives, please refer to the supplementary material for a full description.

**Syntax trees.** A derivation of a given sequence in our grammar can be represented with a *syntax tree*, where the leafs are the terminal symbols that appear in the sequence, and their ancestors are non-terminal symbols. An example of a sequence for a CAD sketch and its syntax tree are shown in Figure 1. The syntax tree provides additional information about a token sequence that we can use to 1) interpret a given token sequence in order to convert it into a sketch, 2) enforce syntactic rules during generation to ensure generated sequences are well-formed, and 3) help our generative model interpret previously generated tokens, thereby improving its performance. Given a syntax tree $T$, we create two additional sequences $Q^3$ and $Q^4$. These sequences contain the ancestors of each token from two specific levels of the syntax tree. $Q^x$ contains the ancestors of each token at depth $x$: $Q^x = (a_T^x(q_1), a_T^x(q_2), \dots)$, where $a_T^x(q)$ is a function that returns the ancestor of $q$ at level $x$ of the syntax tree $T$, or a filler token if $q$ does not have such an ancestor. Level 3 of the syntax tree contains non-terminal symbols corresponding to primitive or constraint types, such as *point*, *line*, or *coincident*, while level 4 contains parameter types, such as *location* and *direction*. The two sequences $Q^3$ and $Q^4$ are used alongside $Q$ as additional input to our generative model.

**Parsing sketches.** To parse a sketch into a sequence of tokens that follows our grammar, we iterate through primitives first and then constraints. For each, we create a corresponding sequence of tokens using derivations in our grammar, choosing production rules based on the type of primitive/constraint and filling in the terminal symbols with the parameters of the primitive/constraint. Concatenating the resulting per-primitive and per-constraint sequences separated by tokens $\Lambda$ gives us the full sequence $Q$ for the sketch. During the primitive and constraint derivations, we also store the parent and grandparent non-terminal symbols of each token, giving us sequences $Q^3$ and $Q^4$. The primitives and constraints can be sorted by a variety of criteria. In our experiments, we use the order in which the primitives were drawn by the designer of the sketch [27]. In this order, the most constrained primitives typically occur earlier in the sequence. The constraints are arranged based on prevalence in the dataset, constraints that are used more frequently in the dataset occur earlier.

## 5 Models

Following [21], we decompose our graph generation problem into two parts:

$$p(S) = \underbrace{p(\mathcal{P})}_{\substack{\text{Primitive} \\ \text{Model}}} \underbrace{p(\mathcal{C}|\mathcal{P})}_{\substack{\text{Constraint} \\ \text{Model}}}$$

Both of these models are trained by teacher forcing with a Cross-Entropy loss. We now describe each of the models.

Figure 2: Grammar of the CAD sketch language. Each sentence represents a syntactically valid sketch. The full grammar is given in the supplementary material.

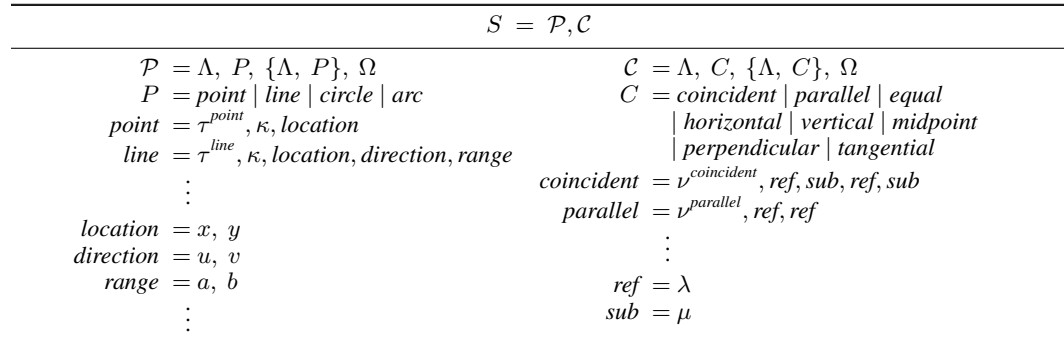

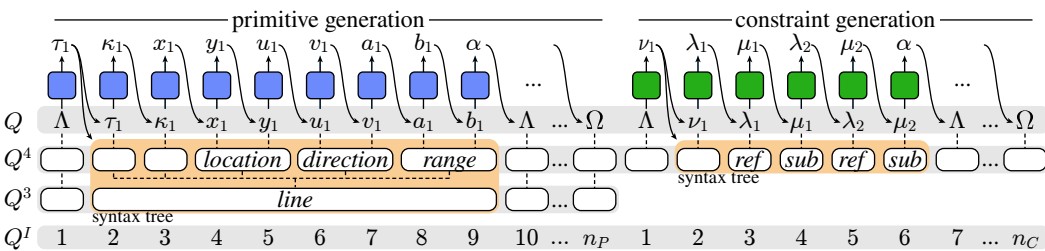

$$S = \mathcal{P}, \mathcal{C}$$

| | |
|---|---|
| $\mathcal{P} = \Lambda,\ P,\ \{\Lambda,\ P\},\ \Omega$ | $\mathcal{C} = \Lambda,\ C,\ \{\Lambda,\ C\},\ \Omega$ |
| $P = \mathit{point} \mid \mathit{line} \mid \mathit{circle} \mid \mathit{arc}$ | $C = \mathit{coincident} \mid \mathit{parallel} \mid \mathit{equal}$ $\mid \mathit{horizontal} \mid \mathit{vertical} \mid \mathit{midpoint}$ $\mid \mathit{perpendicular} \mid \mathit{tangential}$ |
| $\mathit{point} = \tau^{\mathit{point}}, \kappa, \mathit{location}$ | $\mathit{coincident} = \nu^{\mathit{coincident}}, \mathit{ref}, \mathit{sub}, \mathit{ref}, \mathit{sub}$ |
| $\mathit{line} = \tau^{\mathit{line}}, \kappa, \mathit{location}, \mathit{direction}, \mathit{range}$ | $\mathit{parallel} = \nu^{\mathit{parallel}}, \mathit{ref}, \mathit{ref}$ |
| $\vdots$ | $\vdots$ |
| $\mathit{location} = x,\ y$ | $\mathit{ref} = \lambda$ |
| $\mathit{direction} = u,\ v$ | $\mathit{sub} = \mu$ |
| $\mathit{range} = a,\ b$ | |
| $\vdots$ | |

Figure 3: Sequence generation approach. The sequence $Q$ from $1 \ldots n_P$ describes the primitives and from $n_P + 1 \ldots n_C$ the constraints of a sketch. We use two separate generators for the two sub-sequences (blue for primitives, green for constraints). The sequences $Q^4$ and $Q^3$ describe part of the syntax tree of $Q$ and are used as additional input.

### 5.1 Primitive Generator

**Quantization.** Most of the primitive parameters are continuous and have different value distributions. For example, locations are more likely to occur near the origin and their distribution tapers off outwards, while there is a bias towards axis alignment in direction parameters. To account for these differences, we quantize each continuous parameter type (*location*, *direction*, *range*, *radius*, *rotation*) separately. Due to the non-uniform distribution of parameter values in each parameter type, a uniform quantization would be wasteful. Instead, we find the quantization bins using $k$-means clustering of all parameter values of a given parameter type in the dataset, with $k = 256$.

**Input encoding.** In addition the the three inputs sequences $Q$, $Q^3$, and $Q^4$ described in Section 4, we use a fourth sequence $Q^I$ of token indices that provides information about the global position in the sequence. Figure 3 gives an overview of the input sequences and the generation process. We use four different learned embeddings for the input tokens, one for each sequence, that we sum up to obtain an input feature:

$$f_i = \xi_{q_i} + \xi^3_{q^3_i} + \xi^4_{q^4_i} + \xi^I_{q^I_i}, \tag{1}$$

where $q^*_i \in Q^*$ and $\xi^*$ are learned dictionaries that are trained together with the generator.

**Sequence generation.** We use a transformer decoder network [31] as generator. As an autoregressive model, it decomposes the joint probability $p(Q)$ of a sequence $Q$ into a product of conditional probabilities: $p(Q) = \prod_n p(q_n | q_{<n})$. In our case, the probabilities are conditioned on the input features $p(q_n|q_{<n}) = p(q_n|f_{<n})$ where $f_{<i}$ denotes the sequence of input features up to (excluding) position $i$. Each step applies the network $g^P$ to compute the probability distribution over all discrete values for the next token $q_i$:

$$p(q_i \mid f_{<i}) = g^P(f_{<i}). \tag{2}$$

At training time all input sequences are obtained from the ground truth. At inference time, the sequence $Q$ is sampled from the output probabilities $p(q_i \mid f_{<i})$ using nucleus sampling, and sequences $Q^3$ and $Q^4$ are constructed on the fly based on the generated primitive type $\tau$, as shown in

Figure 3. In addition to providing guidance to the network, the syntax described in $Q^3$ and $Q^4$ allows us to limit generated token values to a valid range. For example, we correct the generated values for tokens $\Lambda$ and $\Omega$ to the expected special values if a different value has been generated.

## 5.2 Constraint Generator

The constraint generator is implemented as a Pointer Network [32] where each step returns an index into a list of encoded primitives. These indices form the constraint part of the sequence $q_{>n_p}$, where $n_p$ is the number of primitive tokens in $Q$.

**Primitive encoding.** We use the same quantization for parameters described in the previous section, but use a different set of learned embeddings, one for each primitive terminal token. The feature $h'_j$ for primitive $j$ is the sum of the embeddings for its tokens:

$$h''_j = \xi^\tau_{\tau_j} + \xi^\kappa_{\kappa_j} + \xi^x_{x_j} + \xi^y_{y_j} + \xi^u_{u_j} + \xi^v_{v_j} + \xi^r_{r_j} + \xi^c_{c_j} + \xi^a_{a_j} + \xi^b_{b_j}, \tag{3}$$

where $\tau_j$, $\kappa_j$, $\ldots$ are the tokens for primitive $j$. We use a special filler value for tokens that are missing in primitives. We follow the strategy proposed in PolyGen [21] to further encode the context of each primitive into its feature vector using a transformer encoder:

$$h'_j = e(h''_j, H''), \tag{4}$$

where $H''$ is the sequence of all primitive features $h''_j$. The transformer encoder $e$ is trained jointly with the constraint generator.

**Input encoding.** In addition to the sequence $Q$, we use the two sequences $Q^4$ and $Q^I$ as inputs, but do not use $Q^3$ as we did not notice an increase in performance when adding it. The final input feature is then:

$$h_i = h'_{q_i} + \xi^{4C}_{q^4_i} + \xi^{IC}_{q^I_i}, \tag{5}$$

where $h'_{q_i}$ is the feature for the primitive with index $q_i$, and $\xi^{4C}$, $\xi^{IC}$ are learned dictionaries that are trained together with the constraint generator.

**Sequence generation.** Similar to the primitive generator, the constraint generator outputs a probability distribution over the values for the current token in each step: $p(q_i|h_{>n_p<i})$, conditioned on the input features for the previously generated tokens in the constraint sequence, denoted as $h_{>n_p<i}$. Unlike the primitive generator, we follow PointerNetworks [32] in computing the probability as dot product between a the output feature of the generator network and the primitive features:

$$P(q_i = j | q_{>n_p<i}) = h'_j \cdot g^C(h_{>n_p<i}), \tag{6}$$

where $g^C$ is the constraint generator. This effectively gives us a probability distribution over indices into our list of primitive embeddings. Constraints may also reference sub-parts of primitives, such as line endpoints, defined by the $\mu$ tokens of the constraint (see Figure 3). For $\mu$ tokens, the indices into primitive embeddings are interpreted as IDs for the sub-part; each primitive can have up to $4$ sub-parts (see the supplementary for details). At training time all input sequences are obtained from the ground truth. At inference time, the sequence $Q$ is sampled from the output probabilities $p(q_i|h_{>n_p<i})$ using nucleus sampling, and the sequence $Q^4$ is constructed on the fly based on the generated constraint type $\nu$, as shown in Figure 3. Similar to primitive generation, the syntax in $Q^4$ provides additional guidance to the network and allows us to limit generated token values to a valid range.

## 6 Results

We evaluate our approach on two main applications. We experiment with generating sketches from scratch and also demonstrate an application that we call *auto-constraining* sketches, where we infer plausible constraints for existing primitives. We evaluate conditional models in the supplementary.

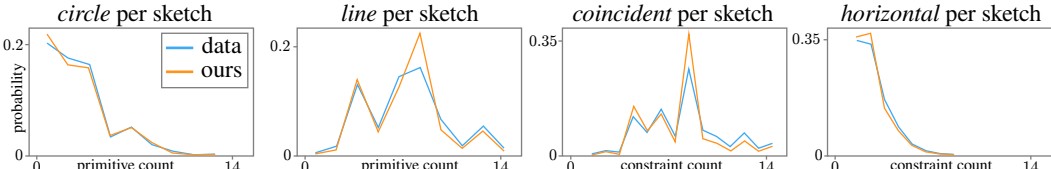

Figure 4: Sketch statistics. We compare the the distribution of *circle* and *line* counts per sketch and the distribution of *coincident* and *horizontal* constraint counts in generated and data sketches.

**Dataset.**    We train and evaluate on the recent Sketchgraphs dataset [27], which contains 15 million real-world CAD sketches (licensed without any usage restriction), obtained from OnShape [2], a web-based CAD modeling platform. These sketches are impressively diverse, however, simple sketches with few degrees of freedom tend to have many near-identical copies in the dataset. These make up $\sim 84\%$ of the dataset and would bias our results significantly. For this reason, we filter out sketches with less than 6 primitives, leaving us with roughly 2.4 million sketches. Additionally we filter sketches with constraint sequences of more than 208 tokens, which are typically over-constrained and constitute $< 0.1\%$ of the sketches. We focus on the 4 most common primitive types and the 8 most common constraint types (see Figure 2 for a full list), and remove any other primitives and constraints from our sketches. We keep aside a random subset of 50k samples as validation set and 86k samples as test set.

**Experimental Setup.**    We implemented our models in PyTorch [24], using GPT-2 [25] like Transformer blocks. For primitive generation, we use 24 blocks, 12 attention heads, an embedding dimension of 528 and a batch size of 544. For constraint generation, the encoder has 22 layers and the pointer network 16 layers. Both have 12 attention heads, an embedding dimension of 264 and use a batch size of 1536. We use the Adam optimizer [12] with a learning rate of 0.0001. Training was performed for 40 epochs on 8 V100 GPUs for the primitive model and for 80 epochs on 8 A100 GPUs for the constraint model. See the supplementary material for more details.

**Baselines.**    We use the sketch generation approach proposed in SketchGraphs [27] as the main baseline, which is based on a graph neural network that operates directly on sketch graphs. Due to the recent publication of the SketchGraphs dataset, to our knowledge this is still the only established baseline for data-driven CAD Sketch generation. This baseline has two variants: *SG-sketch* generates full sketches and *SG-constraint* generates constraints only on a given set of primitives. We re-train both variants on our dataset. Additionally we retrain a DeepSVG [4] model on the SketchGraphs dataset. Details of the retraining for both can be found in the supplementary material. As a lower bound for the generation performance, we also include a *random* baseline, where token values in each step are picked with a uniform random distribution over all possible values. Additionally, for constraint generation, we compare to an auto-constraining method with hand-crafted rules, and as ablation, we compare to variants of our own model that do not use the syntax tree, or only part of the syntax tree as input.

Table 1: Sketch generation. We compare the quality of our learned distribution over sketches to two baselines on the left, and compare three variants of our method using statistics over generated sequences on the right.

<table>
<tr><td colspan="4">(a) Metrics on the test set.</td></tr>
<tr><td></td><td colspan="3">NLL↓ in bits per</td></tr>
<tr><td>method</td><td>sketch</td><td>prim.</td><td>constr.</td></tr>
<tr><td>random</td><td>1020.73</td><td>70.97</td><td>24.14</td></tr>
<tr><td>SG-sketch</td><td>158.90</td><td>-</td><td>2.42</td></tr>
<tr><td>DeepSVG</td><td>100.26</td><td>11.49</td><td>-</td></tr>
<tr><td>ours</td><td>88.22</td><td>8.60</td><td>0.61</td></tr>
</table>

(b) Metrics on generated sequences.

| method | $E_{\text{syntax}}\downarrow$ | $E_{\text{stat}}^{P}\downarrow$ | $E_{\text{stat}}^{C}\downarrow$ | $E_{\text{stat}}\downarrow$ |
|---|---|---|---|---|
| ours ($p = 1.0$) | 19.8 | **0.0058** | **0.0134** | **0.0192** |
| ours ($p = 0.9$) | **18.3** | 0.0185 | 0.0442 | 0.0627 |

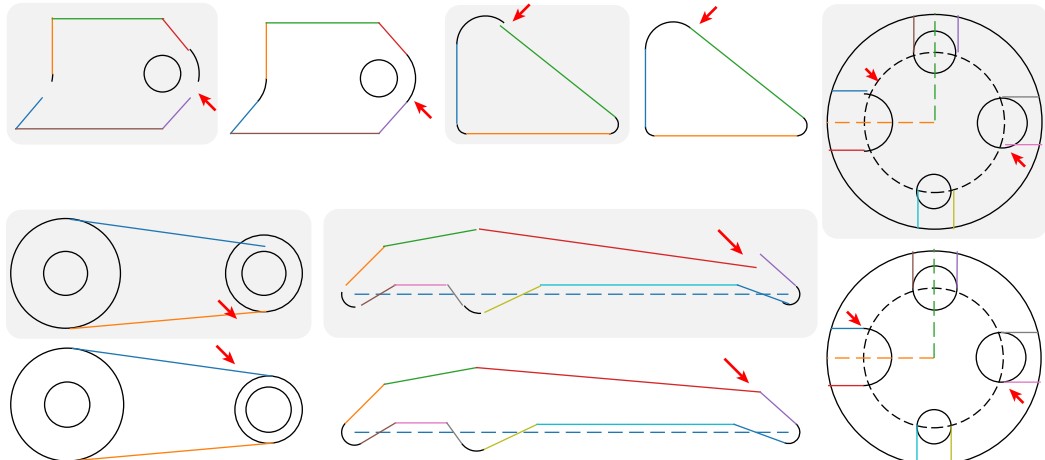

Figure 5: Examples of generated sketches before (gray background) and after optimization to satisfy the generated constraints. Note that the optimization corrects quantization errors (red arrows point out a few changed details). The right-most example is a perturbed testset sketch.

**Sketch Generation Metrics.** We report sketch generation performance using three metrics. The *negative log-likelihood* (NLL) of test set sketches in our models (in bits) measures how much the learned distribution of sketches differs from the test set distribution (lower numbers mean better agreement). In addition to this metric of the generator's performance on the test set, we use two metrics that evaluate the quality of generated sequences. The *syntactic error* ($E_{syntax}$) measures the percentage of sequences with invalid syntax if we do not constrain tokens to be syntactically valid. Lastly, the *statistical error* ($E_{stat}$), measures how much the statistics of generated sketches differ from the ground truth statistics. We compute $E_{stat}$ based on statistics like the number of point primitives per sketch, the distribution of typical line directions, or relative postioning errors, each represented as normalized histogram. $E_{stat}$ is then the earth mover's distance [16] (EMD) between the histograms of generated sketches and test set sketches, see the supplementary material for details. We further split up $E_{stat}$ into $E_{stat}^{P}$ for statistics relating to primitives and $E_{stat}^{C}$ for constraints.

**Sketch Generation Results.** Quantitative results are shown in Table 1. Metrics computed on the test set are shown on the left. We can see that our method leads to the smallest NLL, indicating that the distribution of generated sketches more closely aligns with the test set distribution than both DeepSVG and SG-sketch. Since primitives are constructed using constraints in SG-sketch, we do not show the NLL per primitive for that baseline. DeepSVG can only generate primitives, but as we see in the results, still has lower performance than our model. All three methods perform far better than the upper bound given by the random baseline.

On the right, metrics are computed on 15k generated sequences. We compare two different variants of our results, using two different nucleus sampling parameters $p$. Nucleus sampling clips the tails of the generated distribution, which tend to have lower-quality samples. Thus, without nucleus sampling ($p = 1.0$) we see an increase in the syntactic error, due to the lower quality samples in the tail, but a decrease in the statistical error, since, without the clipped tails, the distribution more accurately resembles the data distribution. We show a few of the statistics we used to compute $E_{stat}$ in Figure 4. We can verify that our generated distributions closely align with the data distribution. Additional statistics are shown in the supplementary material.

The generated constraints can be used to correct errors in the primitive parameters that may arise, for example, due to quantization. In Figure 5, all sketches except the right-most sketch are examples of generated sketches before and after optimizing to satisfy the generated constraints, using the constraint solver provided by OnShape [2]. We can see that our our generated sketches are visually plausible and that the constraint generator finds a plausible set of constraints, that correctly closes gaps between adjacent line endpoints, among other misalignments.

**Auto-constraining Sketches.** Another potential application of our model is to predict a plausible set of constraints for a given set of primitives, for example to expose only useful degrees of freedom

Table 2: Auto-constraining sketches. We compare the quality of our learned distribution over constraints to two baselines on the left, and compare three variants of our method using various metrics over the generated constraint sequences on the right.

(a) Metrics on the test set.

| method | NLL$\downarrow$ in bits per | |
| --- | --- | --- |
| | seq. | constr. |
| random | 259.40 | 24.14 |
| SG-constraint | 19.29 | 1.22 |
| ours | **8.28** | **0.61** |

(b) Metrics on generated sequences.

| method | $E_{syntax}\downarrow$ | $E_{\text{stat}}^{C}\downarrow$ |
| --- | --- | --- |
| ours ($p = 1.0$) | 1.56 | **0.0415** |
| ours ($p = 0.9$) | **1.29** | 0.0442 |

Table 3: Ablation on the primitive model.

| method | NLL/seq.$\downarrow$ | NLL/prim.$\downarrow$ | NLL/token$\downarrow$ |
| --- | --- | --- | --- |
| ours | **79.94** | **8.600** | **0.979** |
| ours -$Q^3$ | 79.96 | 8.605 | 0.980 |
| ours -$Q^3$ -$Q^4$ | 80.35 | 8.640 | 0.984 |

Table 4: Ablation on the constraint model.

| method | NLL/seq.$\downarrow$ | NLL/constr.$\downarrow$ | NLL/token$\downarrow$ |
| --- | --- | --- | --- |
| ours | **8.28** | **0.610** | **0.110** |
| ours -$Q^4$ | 8.47 | 0.633 | 0.113 |
| ours `-shared` | 8.45 | 0.627 | 0.112 |

for editing the sketch, or to correct slight misalignments in the sketch. To evaluate this application, we separately measure the constraint generation performance given a set of primitives, using the metrics described previously.

Quantitative results are shown in Table 2. On the left, we see similar results for constraint generation as for full sketch generation: our learned distribution over constraint sequences is significantly closer to the dataset distribution (lower NLL) than SG-constraint, and both are far from the upper bound given by the random baseline. On the right, we compute constraints for the primitives of $15k$ previously generated sketches, using two different nucleus sampling parameters $p$. Similar to the previous section, deactivating nucleus sampling results in an increase of the syntactic error, but a decrease in the statistical error.

Constraints can also be generated for primitives that do not come from the primitive generator. We perturb the primitives in all test set sketches, generate constraints, and optimize the primitives to satisfy the generated constraints using the OnShape optimizer. Since we have ground truth constraints for the test set, we can measure the accuracy of the generated constraints. The average accuracy on the test set is $98.4\%$. An example is shown in the right-most shape of Figure 5, where the left version (gray background) is the perturbed test set shape and the right version the same shape after optimization. Additional results are shown in the supplementary material.

**Ablation.** We ablate the primitive and constraint models separately. We aim to show that our syntax provides prior knowledge about the the otherwise ambiguous structure of a sketch sequence that improves the generative performance of our models. In Table 3, we show the result of removing sequences $Q^3$ and/or $Q^4$, which contain information about the sequence syntax, from the input of the primitive generator. Removing only $Q^3$ results in a slight performance degradation, while removing both results in a more significant drop. In Table 4, we see that removing $Q^4$ also causes a significant performance drop in the constraint generator. Additionally, we show the importance of grouping the tokens by parameter type in the shared embeddings of the primitive encoder (see Eq. 3). Mixing tokens with different parameter types in the shared embeddings, denoted as ours `-shared`, causes a significant drop in performance.

## 7 Conclusion

In this work, we improve upon the state of the art in CAD sketch generation through the use of transformers coupled with a carefully designed sketch language and an explicit use of its syntax. The SketchGen framework enables the full generation of CAD sketches, including primitives and constraints, or auto-constraining existing sketches by augmenting them with generated constraints.

We left a few limitations for future work. First, we chose only the most common types of primitives and constraints in the dataset to avoid learning from the long tail of the dataset distribution. We can

easily incorporate more types by extending our grammar, and in future work it would be interesting to experiment with using more complex and less common types, such as Bezier curves and distance constraints. Second, the autoregressive nature of our model prevents correcting errors in earlier parts of the sequence and we would like to explore backtracking to correct these errors in future work.

We are excited about future research into generating complex parametric geometry and in examining the explicit use of formal languages and generative language models.

## 8 Broader Impact

There are no foreseeable societal impacts specific to our method. There are societal impacts of generative modeling, machine learning, and deep learning in general that are shared by papers in these areas. The discussion of these broader topics is beyond the scope of this paper.

## 9 Acknowledgements

This work was suppported in part by ARL grant W911NF2120104, and a Vannevar Bush Faculty Fellowship. We would like to acknowledge gifts from Adobe, Autodesk and the UCL AI Centre. We thank the KAUST Supercomputing Lab (KSL) for providing compute infrastructure. Finally, we thank the anonymous reviewers for their constructive comments.

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
