# SketchGen: Generating Constrained CAD Sketches
# Supplementary Material

**Wamiq Reyaz Para**[1]     **Shariq Farooq Bhat** [1]     **Paul Guerrero**[2]

**Tom Kelly**[3]     **Niloy Mitra**[2,4]     **Leonidas Guibas**[5]     **Peter Wonka**[1]

[1] KAUST     [2] Adobe Research     [3] University of Leeds
[4] University College London     [5] Stanford University

## Abstract

In this document we provide additional details and experiments. We give a description of the primitives and constraints used, define the full grammar of our sequences and the key notation used in the paper. A description of the complete experimental setup and the metrics used to measure the quality of different sampling strategies is also provided. Additionally, we provide more quantitative and qualitative results for our sampled sketches and provide a few examples of failure cases. We also include additional experiments with conditional versions of our model, that are conditioned on sketch images.

## 1   Primitives and Constraints

When designing a sketch in a CAD package, a user is typically given access to a set of primitives from which they can create complex shapes. While this set of primitives may differ from package to package, certain primitives are ubiquitous - points, lines, circles, and arcs. Most people are familiar with their properties and can manipulate them easily and intuitively. These packages then create more complex but commonly used building blocks - rectangles, rounded rectangles, etc, from these simpler primitives.

In addition to primitives, constraints are often used in CAD packages to define geometric relationships between primitives that should be maintained during the design exploration process, for example *coincident* constraints on line endpoints to form a longer polyline. These constraints may refer to either to sub-parts of a primitive, as in the polyline example above, or the whole primitive, like a *vertical* constraint on a line.

A complete list of primitive types and constraint types we use in our experiments are shown in Tables 1, 2, and 3. For primitives, we provide all sub-parts that can be target of a constraint, all parameters and an example use case in Table 1. For constraints, we provide all parameters, a short description and an example use case in Tables 2 and 3.

In addition to the primitives we use, CAD packages often include additional primitives like splines. In the SketchGraphs dataset [7], splines only occur in a small percentage ($< 3\%$) of the sketches, making it hard to accurately learn them from data. For this reason, we defer them to future work.

## 2   Complete Grammar

In the interest of space, we only included part of the grammar in the main paper. The complete grammar we use is shown in Figure 1.

35th Conference on Neural Information Processing Systems (NeurIPS 2021), Sydney, Australia.

Table 1: A list of primitives that we support and a visualization to describe their semantics. We use the parameterization used in OnShape [1], where primitives are often over-parameterized to facilitate editing operations. A *line*, is defined by a point $x, y$, a direction $u, v$ from that point, and an interval $a, b$, describing the length of the line segment in the positive and negative direction $u, v$, starting from the point. An *arc* is defined by its center $x, y$, a radius $r$, a direction $u, v$, and an interval $a, b$ describing the angular extent of the arc in the clockwise/counterclockwise direction, starting from $u, v$. The parameter $c$ is an indicator used to flip the direction of $a, b$. A *circle* is defined in the same way, but without the interval $a, b$, which is implicitly assumed to cover the whole circle.

| Primitive | Sub-references | Parameters | Visualization |
|---|---|---|---|
| *point* | Whole | $x, y$ |  |
| *line* | Start End Whole | $x, y, u, v, a, b$ |  |
| *arc* | Start End Center Whole | $x, y, u, v, r, c, a, b$ |  |
| *circle* | Center Whole | $x, y, u, v, r, c$ |  |

Table 2: A list of constraints that we support and a visualization to describe their semantics. A point is parameterized by its coordinates $(x, y)$. A line is parameterized by a point on the line $(x, y)$ (that doesn't have to be the midpoint), a direction $(u, v)$ and two scalar parameters $(a, b)$ to describe how far the two end points are from the midpoint along the given direction. (Continued in Table 3.)

| Constraint | Parameters | Description | Visualization |
|---|---|---|---|
| *coincident* | $\lambda_1, \mu_1, \lambda_2, \mu_2$ | Makes two points coincident. The points can be sub-references of primitives. |  |
| *horizontal* | $\lambda_1, \mu_1, \lambda_2, \mu_2$ | Imposes a horizontal constraint on one/two primitives. |  |

Table 3: (Continued) A list of supported constraints and a visualization to describe their semantics.

| Constraint | Parameters | Description | Visualization |
|---|---|---|---|
| *vertical* | $\lambda_1, \mu_1, \lambda_2, \mu_2$ | Imposes a vertical constraint on one/two primitives. | Line1, Whole, Line1, Whole 
 Line1, Start, Line2, End |
| *parallel* | $\lambda_1, \lambda_2$ | Constrains two lines to be parallel. | Line1, Line2 
 Line1, Line2 |
| *perpendicular* | $\lambda_1, \lambda_2$ | Constrains two lines to be perpendicular.. | Line1, Line2 
 Line1, Line2 |
| *midpoint* | $\lambda_1, \mu_1, \lambda_2$ | Makes one primitive's sub-reference the midpoint of another. | Circle, Center, Line1 
 Point1, Whole, Line1 |
| *equal* | $\lambda_1, \lambda_2$ | Makes one primitive's parameters equal to another's. | Line1, Line2 
 Arc1, Arc2 |
| *tangent* | $\lambda_1, \lambda_2$ | Constrains a line and a circle/arc to be tangential. | Line1, Circle1 
 Arc1, Line1 |

Figure 1: Full CAD sketch language grammar. Each sentence represents a syntactically valid sketch.

$$S = \mathcal{P}, \mathcal{C}$$

$$\mathcal{P} = \Lambda, P, \{\Lambda, P\}, \Omega$$
$$P = point \mid line \mid circle \mid arc$$

$$point = \tau^{point}, \kappa, location$$
$$line = \tau^{line}, \kappa, location, direction, range$$
$$circle = \tau^{circle}, \kappa, location, direction, radius, rotation$$
$$arc = \tau^{arc}, \kappa, location, direction, radius, rotation, range$$

$$location = x, y$$
$$direction = u, v$$
$$range = a, b$$
$$radius = r$$
$$rotation = c$$

$$\mathcal{C} = \Lambda, C, \{\Lambda, C\}, \Omega$$
$$C = coincident \mid parallel \mid equal$$
$$\mid horizontal \mid vertical \mid midpoint$$
$$\mid perpendicular \mid tangential$$

$$coincident = \nu^{coincident}, ref, sub, ref, sub$$
$$parallel = \nu^{parallel}, ref, ref$$
$$equal = \nu^{equal}, ref, ref$$
$$horizontal = \nu^{horizontal}, ref, sub, ref, sub$$
$$vertical = \nu^{vertical}, ref, sub, ref, sub$$
$$midpoint = \nu^{midpoint}, ref, sub, ref$$
$$perpendicular = \nu^{perpendicular}, ref, ref$$
$$tangential = \nu^{tangential}, ref, ref$$

$$ref = \lambda$$
$$sub = \mu$$

# 3 Notation

The full list of key notation in our paper is given in Table 4.

Table 4: Summary of key notation used in the paper.

| Symbol | Description |
|---|---|
| $Q$ | The sequence of tokens representing our sketches. |
| $Q^x$ | The sequence of tokens from level $x$ of the syntax tree. |
| $Q^3$ | The sequence of tokens describing the primitive type (*line*, *point*, . . . ) for each token in $Q$ |
| $Q^4$ | The sequence of tokens describing the parameter type (*location*, *ref*, . . . ) for each token in $Q$ |
| $\Lambda$ | Token marking the start of a new primitive in the sequence $Q$. |
| $\Omega$ | Token marking the end of the primitive and constraint sub-sequences of $Q$. |
| $\tau$ | Token describing the type of primitive. All following tokens up to the next $\Lambda$ are considered part of the primitive. |
| $\nu$ | Token describing the type of constraint. All following tokens up to the next $\Lambda$ are considered part of the constraint. |
| $\kappa$ | Token indicating if a primitive is a construction aid. |
| $\lambda$ | Token corresponding to a reference (and index) into a list of primitives. |
| $\mu$ | Token describing the sub-reference type (`Start`, `End`, `Center`, `Whole`) |
| $g^P(\cdot)$ | The primitive model. Modeled as a **masked** Transformer decoder. |
| $e(\cdot, \cdot)$ | (Constraint model). The primitive encoder of the constraint model. Modeled as an **unmasked** Transformer encoder |
| $g^C(\cdot)$ | (Constraint model). The pointer generator for the constraint model. Modeled as a **masked** Transformer decoder |
| $\xi$ | Learnable embedding |
| $f/h$ | Features for primitives/constraints. |

# 4 Detailed Experimental Setup

## 4.1 Architecture

Our architectural building blocks are Transformer [9] blocks with self-attention [2], stacked in multiple layers. We used GPT-2 [5] style blocks with code based on the HuggingFace [11] library as the starting point. The primitive model is then a standard Transformer decoder with autoregressive masking. The constraint model is a Pointer-Network [10] with the pointer model masked autoregressively and an unmasked encoder. We add support for our embedding strategy and our additional input sequences.

## 4.2 Training

For training all our models, we used a constant learning schedule, with no weight decay. We used Dropout [8] with a probability of $0.2$ after calculating all embeddings, and within each attention head. The constraint model was trained with gradient clipping to a norm of $1.0$ to improve training stability. We trained the Primitive Model for 40 epochs, and the constraint model for 80 epochs. The primitive model took about 30 hours to train, while the constraint model took about 16 hours. For both models, we use early stopping on the validation dataset, using the model with the best performance on the validation set.

## 4.3 Setup for the SketchGraphs Baseline

We re-train the SketchGraphs [7] baseline on our subset of the SketchGraphs dataset using code obtained from the author's GitHub repository[1]. We try to follow their training scheme as closely as possible and make only a few changes to improve training times - for the complete generative model, SG-sketch, we double the batch size from 8192 to 16384 and use 4 GPUs instead of 3.

We use the same subset of constraints that we use in our method to train the autoconstraint model, SG-constraint. For the complete generative model SG-sketch however, we need to use all constraints that are available in the dataset for a fair comparison. The SketchGraphs baseline does not directly generate primitive parameters, it only generates a set of constraints on the parameters, that need to be solved to obtain the actual primitive parameters. Without the full set of constraints, the primitive parameters would be underdetermined, i.e. not well-defined after solving the constraints.

## 4.4 Setup for the DeepSVG Baseline

DeepSVG is designed to be a generative model for SVGs. Consequently it models only primitives, not constraints. SVGs are composed of mulitple *paths* each of which itself is composed of multiple commands. In order to train DeepSVG with our encoding, we map a single primitive to a single *path*, and each path then only consists of a single *command* describing the primitive. We obtain code from the official repository[2], and train on 4 Nvidia RTX Titan GPUs with a batch size of $256$, and a learning rate of $4 \times 10^{-3}$. We set the maximum number of paths to $16$, the number of arguments to $9$ and the number of commands to be $6$, including $4$ commands for our primitive types, and $2$ commands for the special `<SOS>` and `<EOS>` tokens that are needed by DeepSVG. Please refer to DeepSVG [3] for details on their encoding scheme. For all other hyperparameters, we use the original values set by the authors.

In order to report the NLL, we use the strategy used by DeepSVG and add up the log-likelihood contributions from the *visibility* logits, *command* logits and the *argument* logits. DeepSVG uses a fixed length sequence for *arguments*. This can lead to many arguments being unused for a given *command*. In such cases, we do not consider the contribution of unused arguments to the NLL, and mask those arguments out. The same procedure is applied during training, as is default for DeepSVG.

# 5 Metrics: $E_{\text{syntax}}$ and $E_{\text{stat}}$

We describe the two metrics $E_{\text{syntax}}$ and $E_{\text{stat}}$ in more detail.

---

[1]https://github.com/PrincetonLIPS/SketchGraphs
[2]https://github.com/alexandre01/deepsvg

**Esyntax**  is a measure of how well our model respects our grammar. The grammar gives us the length of the subsequence that describes a single primitive or constraint. As an example, a *point* is described by a subsequence of length 3: $\tau^{point}, x, y$. The $\Lambda$ token separates primitive subsequences, so if the model makes an error in the length of any given subsequence, it will not correctly predict the position of the $\Lambda$ token that marks the start of a new subsequence. At inference time, our grammar allows us to infer the correct position of the $\Lambda$ tokens, so we can correct these errors by forcing the network to produce a $\Lambda$ at the correct position. $E_{\text{syntax}}$ measures how often we need to intervene and force a $\Lambda$ token where the network would have sampled a different token, as a percentage of the total number of $\Lambda$ tokens in a sequence:

$$E_{\text{syntax}} = \frac{\#(\text{correctly sampled } \Lambda \text{ tokens})}{\#(\Lambda \text{ tokens expected from grammar})} \times 100 \tag{1}$$

**Estat**  compares statistics of generated sketches to statistics of ground truth sketches.

We sample a large number of sketches, and record the distributions of various sketch properties like line lengths, and number of primitives/constraints per instance. After normalizing the domain of these distributions to the unit interval $[0, 1]$, we can compare the distributions for generated sketches to the distributions for ground truth sketches using the Earth Mover's Distance[3] (EMD) [6].

We record statistics over several different groups of sketch properties. For primitives, we record the following groups:

- **Cardinality**: the distribution of primitive counts per instance for each type of primitive.
- **Position**: the distribution of $x, y$ parameter values for each type of primitive.
- **Size**: the distribution of *line* lengths, *arc* lengths, and *circle* radii.

We average the EMDs of all statistics inside a group to get a statistical error per group, and then average these errors across groups to get the statistical error for primitives $E_{\text{stat}}^{P}$. For constraints, we record:

- **Cardinality**: the distribution of constraint counts per instance for each type of constraint.
- **Tangent**: the distribution of absolute radius differences between *circle* or *arc* primitives that are connected by a *tangent* constraint.
- **Perpendicular**: the distribution of absolute length differences between *line* primitives that are connected by a *perpendicular* constraint.
- **Horizontal**: the distribution of absolute length differences between *line* primitives that are connected by a *horizontal* constraint.
- **Vertical**: the distribution of absolute length differences between *line* primitives that are connected by a *vertical* constraint.
- **Coincident**: the distribution of absolute length differences between *line* primitives that are connected by a *coincident* constraint.

Similar to the primitive statistics, we first average the EMDs inside a group and then across groups to get the statistical error for constraints $E_{\text{stat}}^{C}$. The full statistical error $E_{\text{stat}}$ is the sum of the primitive and constraint errors: $E_{\text{stat}} = E_{\text{stat}}^{P} + E_{\text{stat}}^{C}$.

Note that this is only a sample of all possible statistics over primitive parameters, and pairwise relationships between primitives, but as we see in Tables 1 (b) and 2 (b) of the main paper, they correspond well to expected behavior - nucleus sampling with $p = 1.0$ samples well from the tail of the distribution and both $E_{\text{stat}}^{P}$ and $E_{\text{stat}}^{C}$ are correspondingly lower when compared to nucleus sampling with $p = 0.9$.

## 6 Additional Quantitative Results

We show additional examples of the statistics we use to compute our metrics $E_{\text{stat}}$.

---

[3]We use the implementation in `scipy.stats.wasserstein_distance`

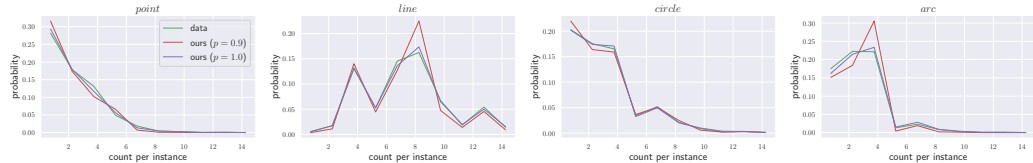

Figure 2: Sketch statistics for primitives: we compare the the distribution of primitive counts per sketch in generated and data sketches.

In Figure 2, we show the distribution of the number of primitives per instance for all primitive types, and in Figure 3, we show the distribution of the number of constraints per instance, for all constraint types. Note that our model infers constraints similar to the dataset, which in turn comes from human designers.

Constraints can be used to limit the degrees of freedom (DoF) of a sketch, for example to expose only DoF that are useful for a given editing task. In the inset figure, we examine how many DoF our constraints remove per sketch, and compare to the number of DoF removed by the ground truth constraints. We see that our model is in very close agreement with the constraints imposed by a human designer.

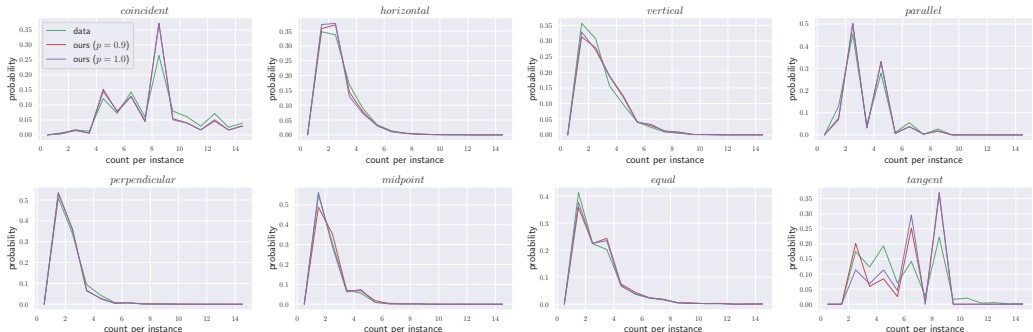

Figure 3: Sketch statistics for constraints: we compare the the distribution of constraint counts per sketch in generated and data sketches.

# 7 Comparison to Hand-crafted Rules for Constraints

Some CAD packages have rule-based systems to suggest constraints to the user. As an example, the package might 'snap' line endpoints by imposing a *coincident* constraint on line endpoints that are in close proximity, or a *parallel* constraint may be suggested for lines that are approximately parallel.

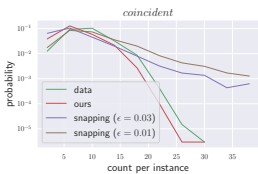

We compare our approach to a baseline where coincidence constraints are created for any pair of points or line endpoints that are within a distance $\epsilon$ of each other. In the inset figure, we compare the number of coincident constraints created by this baseline to our method and to the ground truth. We see that the snapping baseline over-estimates the number of coincidences significantly with a snapping threshold of $\epsilon = 0.03$ and $\epsilon = 0.1$, since the lack of context-dependent reasoning results in several false positives.

# 8 Additional Qualitative Results

## 8.1 Primitive Generation

In Figure 4, we show primitives generated by our primitive model. Notice that our quantization introduces inaccuracies in the primitive parameters. For example, line endpoints that *intuitively*

seem like they should be coincident are not fully coincident, or Line-Circle pairs which should be tangential are not. These inaccuracies can be corrected using the generated constraints. Apart from these inaccuracies, the primitives tend to form plausible sketches that also often exhibit symmetries, right angles, parallel lines, etc. as we would expect in a real sketch.

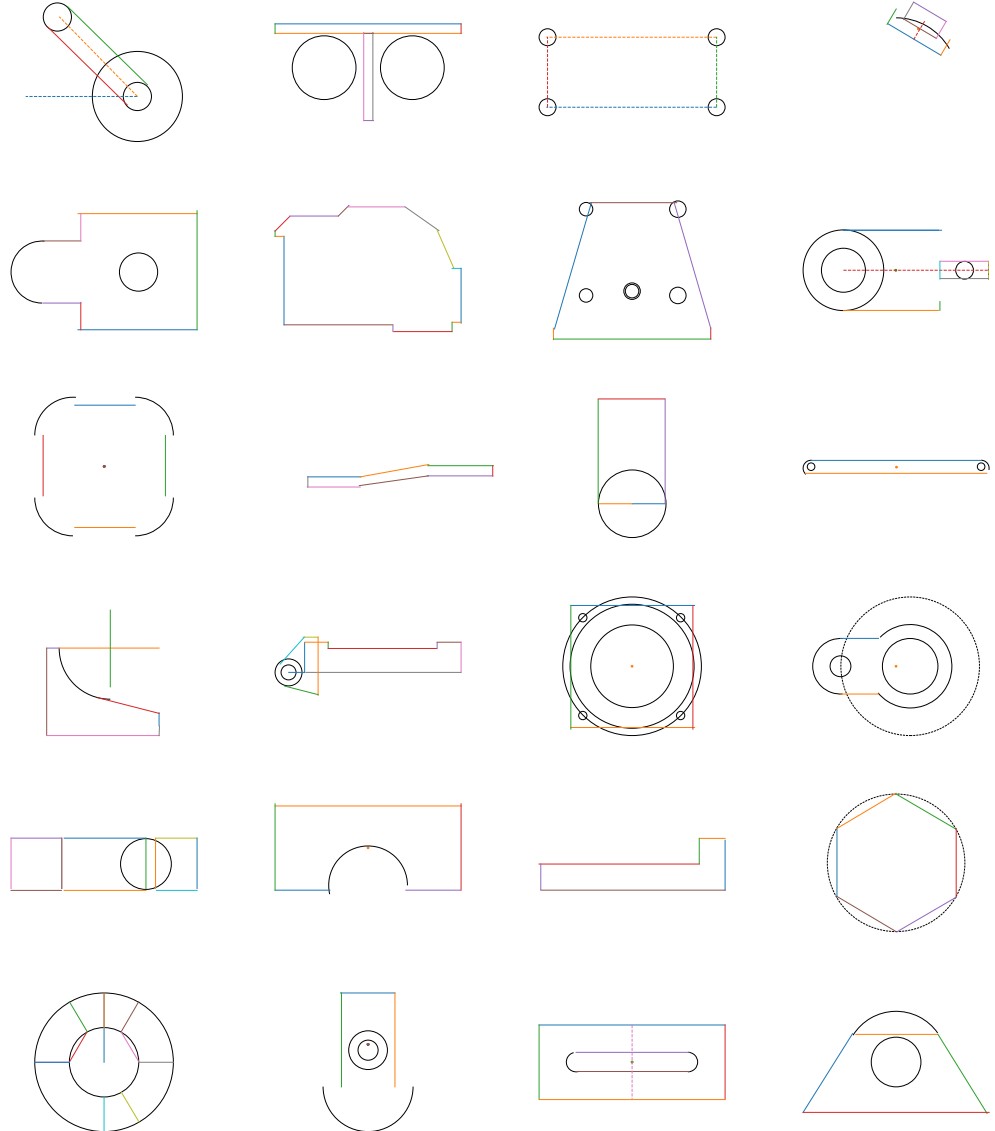

Figure 4: Primitive Generation. We show examples of generated primitive samples. Different colors denote different primitives.

## 8.2   Full Sketch Generation

Generating constraints in addition to primitives allows us to optimize the primitives to satisfy the generated constraints (we use the solver provided by OnShape [1]). Additionally, these constraints can be useful for down-stream tasks, for example they remove unwanted degrees of freedom when exploring variations of a sketch. In Figure 5, we show generated sketches before and after optimization to satisfy the generated primitives. Notice that in many cases errors introduced by quantization or inaccuracies of the primitive generator can be corrected through this optimization.

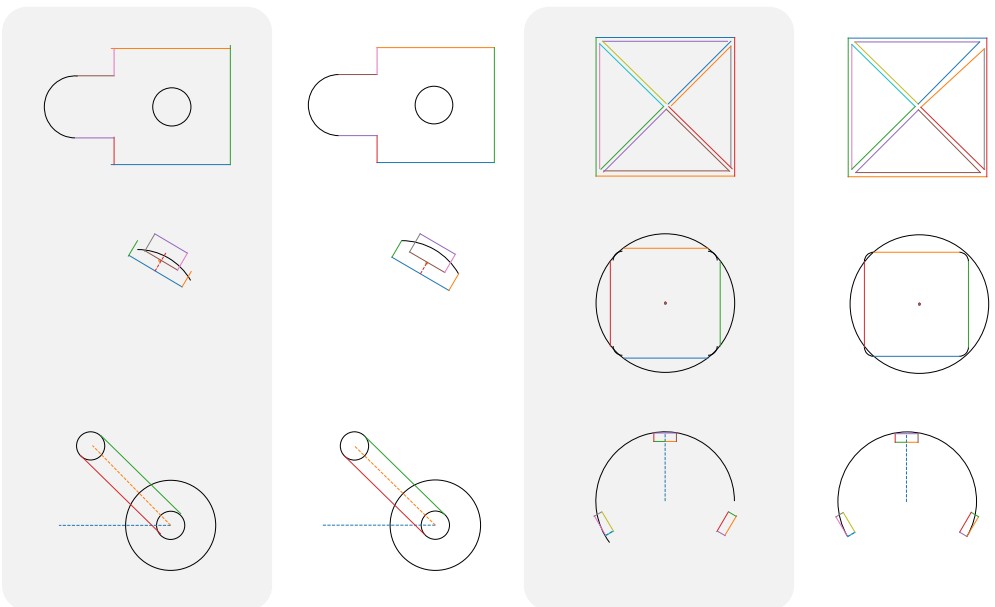

Figure 5: Full sketch generation. We show examples of generated sketches before (gray background) and after optimization to satisfy the generated constraints.

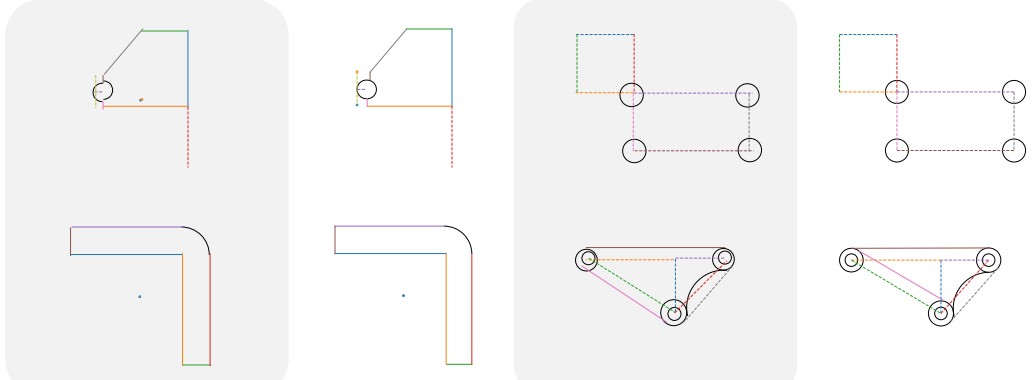

Figure 6: Auto-constraining sketches. We show examples of perturbed test set sketches before (gray background) and after optimization to satisfy generated constraints. The last example shows where constraints might sometimes fail to recover the original sketch - the tangency is maintained on the pink line, but the final tangent lies on the *wrong* side.

## 8.3 Auto-constraining Sketches

We can also generate constraints for existing sketches, effectively inferring the plausible degrees of freedom for a given sketch. We perturb sketches from our test set and infer constraints using our constraint generator. Examples are shown in Figure 6. Notice that in many cases, the generated constraints allow us to improve the alignment of the perturbed primitives.

## 8.4 Failure Cases

We have different kinds of failure modes - the primitive model itself may generate samples that are *out-of-distribution* with errant lines or arcs that do not terminate at another arc or line, a second failure mode is the constraint model generating unsatisfiable constraints like a line end being coincident with two distinct points, finally there may also be cases where the constraints are satisfiable, but do not match the primitives well, resulting in an *out-of-distribution* sketch. Examples of each case are

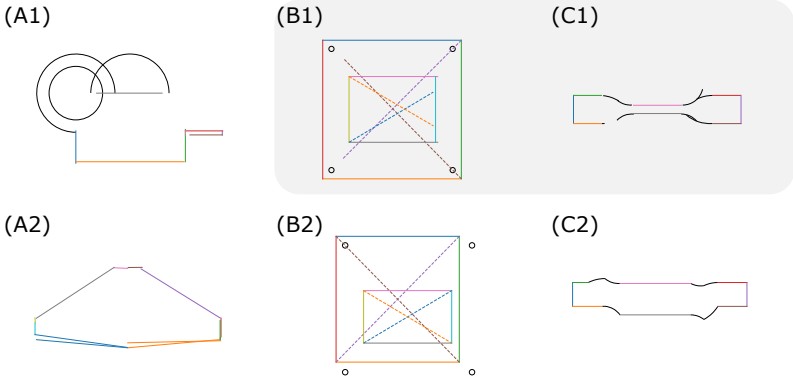

Figure 7: Failure cases. Generated primitives may be *out-of-distribution* (left), for example the sketch may be missing primitives (A1), or have redundant lines (A2). Generated constraints may be missing (center); the small circles, for example, should be constrained to the corners of the big rectangle (B1). Generated constraints may not match the primitives (right); the small arcs, for example, are missing *tangent* constraints, which is why they join at unnatural angles (C2). Grey background indicates sketches before optimization

shown in Figure 7. All of these cases are failures of either the primitive- or the constraint generator and could be improved, for example, by (i) improving the performance of these generators, through architecture improvements or a larger dataset where all primitive and constraint types are represented more evenly, or (ii) introducing additional syntactic or semantic validity checks at inference time, which could be facilitated by our grammar.

# 9    Conditional Models

We also perform experiments with a conditional model. The conditional model predicts a constrained CAD sketch given an image of the sketch, effectively parsing the image by converting it into its constituent primitives. It can also be viewed as a translation task from an image of the sketch to a parametric description of the sketch.

Our conditional model is equipped with a ViT-like [4] encoder which takes in 128x128 images of sketches rendered to emulate a hand-drawn style. A few examples of our style are shown in Figure 8. We use a `Conv-GELU` structure to convert the image into a sequence $\mathcal{I}$ that is 64 elements long. This is achieved by setting the stride and kernel size of the convolution to 16 and the number of output channels to the transformer dimensions. We add a learnable position embedding . This is followed by 8 layers of unmasked GPT-2 blocks. In our primitive model, we add a cross-attention layer that attends to the sequence $\mathcal{I}$. In our constraint model, this cross-attention is performed in the primitive encoder.

## 9.1    Qualitative Results

In Figure 9, we show a few results from our conditional primitive model, applied to test set sketches rendered to an image in a hand-drawn style, as shown in the left-most column. The ground truth primitives are shown in the right-most column. The middle columns show different samples generated by our model for the given sketches. We show multiple different samples per sketch, since the same sketch image can sometimes be achieved with different primitive sequences. For example, a circle may be represented by two arcs or a single circle, and some ambiguity may also arise from the order of primitives in the sequence. In the results we see that our model is able to discover most of the primitives that were used to generate the image with high accuracy. Primitives that function as construction aids are rendered with dotted lines and are correctly identified as construction aids by our conditional model.

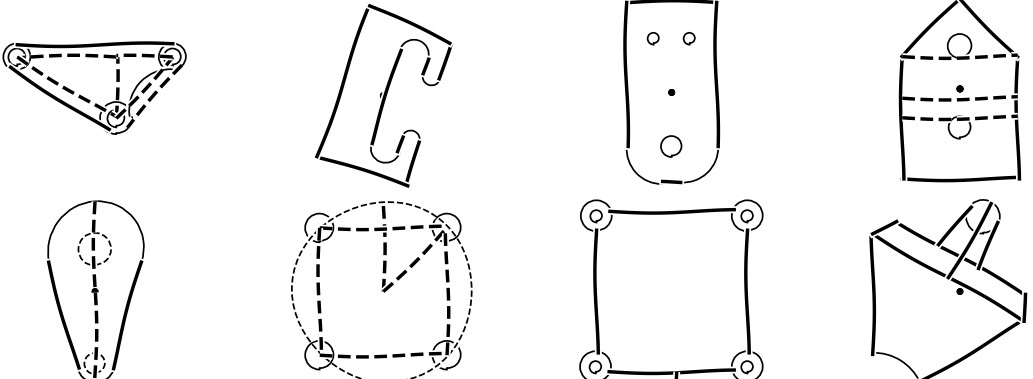

Figure 8: A few representative examples of our rendered sketches. These are used as condition for the conditional model. Construction aids are rendered with dotted lines.

Table 5: Quantitative evaluation on the conditional model.

|  | NLL ↓ in bits per seq. | |
| --- | --- | --- |
|  | unconditional | conditional |
| **Primitive model** | 79.94 | 52.59 |
| **Constraint model** | 8.28 | 7.72 |

## 9.2 Quantitative Results

We measure the effect of adding the image conditioning by recomputing the NLL for the conditional model. The results are shown in Table 5. Using additional supervision in the form of an image reduces the NLL of the model over the dataset. This means that the model is less uncertain about the sequences given an image of the target sketch. This is makes sense intuitively, since an image provides information about the target primitives and their parameters. The effect is weaker for the conditional constraint model. We hypothesize that the constraint model receives a lot less information from the image of the sketch, since constraints are not shown explicitly in the image. This gives the conditional constraint model little additional information over the primitive configuration that it already has access to in the unconditional setting.