# OpenReview forum: "SketchGen: Generating Constrained CAD Sketches"
_NeurIPS.cc/2021/Conference — NeurIPS 2021 Poster_

### Official Review · Reviewer_wERC · 2021-07-12

**Rating:** 7
**Confidence:** 5

**Summary:**

This paper proposes a deep generative model for widely used computer-aided design (CAD) sketches, which are typically composed of parametric primitives (e.g. line, arc) and geometric constraints (e.g. coincidence, parallelism). The problem is challenging because of the complexity of the sketch graph and the heterogeneous nature of the primitives and constraints. To this end, the authors carefully develop a language for CAD sketches to represent each sketch as a sequence of tokens, and then make use of Transformers network to model the data distribution in an autoregressive manner. Evaluation is carried out on unconditional generation and sketch auto-constraining. The generated CAD sketch can be further optimized via an external solver to meet the generated constraints.


**Limitations And Societal Impact:**

Suggestions as above. Mostly, I hope the comparison to DeepSVG can be included.

**Main Review:**

The novelty of this work is clear to me, as a first attempt on generative modeling for engineering CAD sketches (together with some concurrent works), it leverages the similarity between CAD sketches and natural language processing and develops a language to properly represent both parametric primitives and geometric constraints.

The technical part looks sound and there are some interesting design choices, such as the three levels of syntax tree (line 157-158) and non-uniform quantization (line 179-180). Also, the post-processing with an external constraint solver further improves the results,  which is not considered in prior works.

My main concerns are as follows:
1. Although learning a generative model for CAD sketches of engineering standard is interesting, random generation from scratch seems to have less practical importance. I can hardly think of a situation where random CAD sketch generation would be useful. On the other hand, auto-constraining is a more compelling use case, but is less touch upon in the paper. Besides, generation on conditional inputs (e.g. image, partial sketch) is another way to go. It will be better if the author could show some results on that.
2. For the comparison part, it would be better to compare with more previous sketch generation works (e.g. DeepSVG), as the SG-sketch method is not a very strong baseline. I understand that DeepSVG does not model the constraints, but it's still possible to purely compare the sketch quality (using their proposed metrics).
3. All the metrics are in the parametric space. Since the CAD language finally results in a visual 2D sketch, it would be better to include some measurements in the image space, e.g. the image reconstruction error.
4. Another minor point is that the test set seems small, 86K out of 2.4M (3.6%). It might result in much variance during different training runs.

Some confusing details:
1. In line 168-169, what exactly does "the frequency of constraints" refer to? Is it the frequency in the whole dataset or the frequency in the single sketch.
2. How is the token "sub" represented in the constraint generator? By definition, it should indicate some part of a primitive. I'm a bit confused how exactly it is represented as the pointer seems to refer only an entire primitive.



**Time Spent Reviewing:**

5-6 hours

---

> ### Author Response · Authors · 2021-08-10
> **Response to Reviewer wERC**
>
> ### Conditional Generation
> To show a few more practical applications, we will add additional auto-constraining examples and introduce conditional generation from images as additional applications (thanks for the suggestion). Specifically, we use CAD sketches that are rasterized into an image as input and output a sequence of primitives and constraints that reproduces the sketch and has plausible constraints on the primitives. We have already implemented the necessary training and evaluation setup, but did not have time to fully train the experiment in the short time of the rebuttal.
>
> ### Additional Baselines
> As you pointed out, DeepSVG or SketchRNN do not generate heterogeneous primitives or any constraints, thus they cannot model CAD sketches, and do not have the down-stream advantages that CAD sketches provide, like reducing undesired degrees of freedom for easier exploration, easier and more precise editing, etc. Keeping that in mind, we will include a comparison to DeepSVG in the final version. We already made the necessary adjustments to the code of DeepSVG, but did not have enough time to finish running the full experiment in the short time of the rebuttal.
>
> ### Metrics in image space
> Note that we do include some statistical metrics in image space (we need to compare properties of distributions since we don’t have ground truth for individual generated sketches). Specifically, $E_{stat}^C$ measures the positioning errors of primitives that are connected by constraints. We will clarify this in the main paper, as an exact description of $E_{stat}^C$ is currently only available in the supplementary. Additionally, we can provide image-space metrics on individual sketches for the new conditional generation application we described above.
>
> ### Size of the test set
> We believe that the 86k sketches in our test set do a reasonable job of capturing most of the variability in the dataset.
>
> ### Clarifications of details
> “frequency of constraints" refers to how often the constraints occur in the dataset. We will clarify this in the text.
> Regarding the “sub” tokens, each primitive type has a certain number of sub-references that can be targeted by a constraint. The list of sub-references for each primitive is given in Table 1 of the supplementary. The output of the constraint generator for a “sub” token is an index into this list. The index is obtained by comparing the feature generated by the constraint generator to the first four encoded primitives (since maximum number of different sub-references is 4 for any primitive type), i.e. the code of the first four primitives doubles as an encoding of the sub-reference type. In the future, we plan to have a separate set of dictionary codes for sub-references. We will clarify this in Section 5.2.

---

### Official Review · Reviewer_vzeX · 2021-07-16

**Rating:** 6
**Confidence:** 3

**Summary:**

The paper proposes a generative model for CAD sketches. The task of modeling CAD format is challenging due to the heterogeneity of the primitives and the constraints. In other words, the CAD representation lacks a common interface. To deal with this problem, the paper presents a domain-specific language enabling tokenizing a CAD sketch. Being able to represent sketch primitives and constraints allows using sequence models such as Transformers.
The task of modeling CAD sketches is decomposed into the modeling of primitives and the constraints given the set of primitives. By using the predicted constraints, a post-processing optimization can be used to correct the alignment and positioning issues with the predicted primitives.


**Limitations And Societal Impact:**

The limitations of the work are addressed.

**Main Review:**

**Originality**
The paper proposes a domain-specific language and a task decomposition to model the primitives and the constraints separately. Both approaches have been used in different domains and shown to be effective. The paper successfully combines them for CAD sketches. The presented model, training objectives, and evaluations are not novel, but reasonable choices for this task. Modeling primitives and the relations among them is a common pattern followed in “layout modeling” tasks. In the related work section, the authors make connections to relevant works. Though, the long citation list (Ln. 86) is not useful as the “CAD-like representations” statement is too vague to summarize the listed works. [1, 2, 3] should also be included.

**Quality**
- The paper follows a symbolic approach in representing the CAD sketches rather than images. It is well motivated. The proposed grammar and the language disambiguate the tokens (i.e., primitive and constraint parameters) and provide semantic information to the model.

- Due to the quantization of the 2D space, the primitive model often makes misaligned predictions. The results are arguably good as the model requires a post-processing optimization to correct the placement of the primitives with respect to the constraints. The constraint model, on the other hand, performs well in determining the constraints between the primitives (98% constraint prediction accuracy). This enables running the optimization with a high success rate.

- The paper presents results on the quantized data only. Training the primitive model on real-valued data could be an interesting baseline. It could alleviate the issues with positioning due to the quantization. For example, in [2], the Transformer model makes real-valued predictions for stroke positions.

- Both for the primitives and the constraints, the paper follows a sequential approach. Is the ordering of the shapes or constraints carry useful information (see [2])? Running the same model with a set assumption would be a useful ablation.

- How are the $Q^3$ and $Q^4$ sequences processed by the model? Consider the example illustrated in Fig. 3. The “line” label starts at index 2 and continues until index 9. Are the line embeddings (i.e., $\xi_{q_i^3}^3$) the same across these indices?

- The paper provides an interesting model design in the introduction section (Ln. 44-49). Having it among the baseline models would strengthen the story of the paper. Having the syntax labels as part of the padded representation (see my previous comment regarding $Q^3$ and $Q^4$), I do not see a reason for why this baseline fails.

- Ablations: In Table 3, the model is evaluated with and without $Q^3$ and $Q^4$ input sequences. I do not think the difference is significant. What is the difference between NLLs of 79.94 and 79.96?
- Qualitative comparison: The paper does not present qualitative results for the baseline SG-sketch.
- NLL metric for the generated sketches: How is it calculated for the sketches generated from scratch?
- The paper does not provide a measure for the positioning errors. Decomposing it into prediction (i.e., incorrect bin) and quantization errors (i.e., correct bin prediction) could be a useful metric.

[1] Ellis et al., Learning to Infer Graphics Programs from Hand-Drawn Images

[2] Aksan et al., CoSE: Compositional Stroke Embeddings

[3] Wang et al., Deep Convolutional Priors for Indoor Scene Synthesis

Note: I tried to open the manuscript both in Adobe Reader and Chrome browser. There was no gray background in the figures for me. I interpreted the results with failures as the ones with a gray background.

**Clarity**
The paper is well organized. I find the grammar notation cluttered, and hence it is not straightforward to follow section 4. I took several passes. A toy example (like the one in Fig. 1) with numbers along with the symbols could be useful.

**Significance**
It is an interesting and rather new problem for the community. The proposed approach is reasonable, yet it is not evaluated thoroughly due to the lack of self-comparisons. As I mentioned under the **Quality** section, more evidence could be provided to support the design choices. The provided ablations are not very informative.

-------------------
**Update:**
I thank the authors for their response. After reading the other reviews and the rebuttal, I decided not to chance my rating. The paper presents an interesting problem and a technically solid approach. The submission could be stronger by supporting some of the design choices or the claims with experimental evidence.

**Time Spent Reviewing:**

5

---

> ### Author Response · Authors · 2021-08-10
> **Response to Reviewer vzeX**
>
> ### Related work
> Thanks for the additional references. The generation of structured output in our method is indeed related to other methods with structured output such as [2,3], but these have a less heterogeneous structure. We will add these to a few examples we have already listed in the second paragraph of Section 2. Program inference [1] is also an interesting association, although is currently quite limited in its expressiveness and applicability. We will include and discuss these references in Section 2. Currently, our description of the related methods is quite compressed due to space constraints, but we will do our best to expand the description for the list on line 86 without going over the page limit.
>
> ### Ablation with un-quantized data
> Typically quantization is used to allow for arbitrary multi-modal distributions over the quantized bins, which often increases the accuracy of predicted values, as also shown by the majority of generators using quantization. CoSE [2] uses a Gaussian mixture model, which is also multi-modal, but limits the range of possible distributions somewhat. It is hard to judge how accurately CoSE predicts values, since small inaccuracies would be hard to notice in the hand-drawn sketch style used in the results. Inaccuracies are likely to be more noticeable in our precise CAD sketches. Investigating the advantages of quantized prediction, which is used in most current generators based on the transformer architecture, and unquantized prediction would be a significant contribution on its own and seems out of scope for this paper.
>
> ### Ablation with different sequence order
> A consistent ordering of elements has been shown to be beneficial for models that operate on sequences. We use the order in which elements were generated by users, where typically the most constrained primitives are typically drawn first. There might also be alternative orderings that have some consistency, like the scanline order used in PolyGen. However, the order of primitives and constraints is not the focus of our paper, since our syntax trees and active constraints are applicable to any order of primitives and constraints. To keep our paper focused, we would prefer not to add too much detail on the ordering.
>
> ### Processing of $Q^3$ and $Q^4$
> The model gets an embedding of the current token in $Q^3$ and $Q^4$ in each iteration. In Figure 3, your assumption is correct, the model gets the same ‘line’ embedding in $Q^3$ for steps 2-9, while the content of $Q^4$ changes several times in that interval.
>
> ### Padded primitives/constraints as baseline
> Note that our argument in the introduction was that padding primitives/constraints would result in exactly the same problem plus the additional task of generating the padding. The suggestion of using our current solution (including syntax trees and active constraints) plus generating additional padding tokens is thus a valid solution to generate these padded sequences, though nothing would be gained from the added padding tokens.
>
> ### Qualitative results for SketchGraphs baseline
> Thanks for the suggestion, we will add qualitative results of the SketchGraphs baseline to the supplementary material.
>
> ### Benefits of using the sequences $Q^3$ and $Q^4$
> Removing the sequences $Q^3$ and $Q^4$ does have a significant effect on the results. Especially on sketches with complex primitives and constraints, they help regularize the generation problem by providing some prior knowledge about the structure of the sketch representation. We will add a few qualitative results to illustrate this difference.
>
> ### NLL metric computation
> The NLL metric is computed by evaluating sketches from the test set in the generative model and computing how many bits are needed to represent the test set sketch (i.e. how surprised is the generative model by seeing the test set sketch). We will clarify this when introducing the metric.
>
> ### Metrics for the quantization error.
> Note that the metric $E_{stat}^C$ *does* measure the quantization error statistically (we need to compare properties of distributions since we don’t have ground truth for individual generated sketches). Specifically, $E_{stat}^C$ measures the positioning errors of primitives that are connected by constraints. We will clarify this in the main paper, as an exact description of $E_{stat}^C$ is currently only available in the supplementary.
>
> ### Clarity of Section 4
> Thanks for the suggestion, we will improve the clarity of Section 4, possibly with a toy example, or with a clearer grammar notation.

---

### Official Review · Reviewer_Wjpj · 2021-07-16

**Rating:** 6
**Confidence:** 3

**Summary:**

The paper proposes a new CAD sketches generation method, which designs a new language for CAD primitives and constraints and proposes a transformer-based CAD sketch generator. Results on sketch generation and auto-constraining sketches show the effectiveness of the proposed method.


**Limitations And Societal Impact:**

The authors discussed the limitations and failure cases in the paper.

There are not much negative societal impact of this work.

**Main Review:**

Strengths:

1. The paper decomposes CAD sketched into primitives and constraints and designs a new language for CAD sketches. The grammar is carefully designed.

2. The generative model achieves successful sketch generation from scratch and generating constraints for primitives.

3. The quantitative evaluation and ablation studies are sufficient. Ablation studies validate additional syntax tree information is helpful for both primitive and constraint generation.

4. The paper is well-written, and the figures are clear and helpful for understanding.

Weaknesses:

1. The primitive and constraint types are quite limited and can only generate relatively simple sketches.

2. The design of the generative model is similar to PolyGen[9], primitive generator corresponds to vertex model and constraint generator corresponds to face model. This limits the novelty of the model.

Considering the complex CAD sketch language design and limited novelty of the generative model, I regard this paper as marginally above the acceptance threshold.

***
Update:

I thank the authors for their response. The response addressed my concerns about the primitive and constraints types, and the difference between the proposed method and PolyGen.

After reading the other reviewers' comments and author rebuttals, I remain my original rating. The submission solves an interesting and challenging problem, and proposes a solid method to solve it. I recommend acceptance.

**Time Spent Reviewing:**

4

---

> ### Author Response · Authors · 2021-08-10
> **Response to Reviewer Wjpj**
>
> ### Limited novelty compared to PolyGen
> A main challenge of our problem setting is the heterogeneity of the primitives and as well as handling constraints, as also acknowledged by the other reviewers. This is also the main difference compared to PolyGen’s problem setting, which works with homogeneous vertices and edges. This new challenge motivates our new technical contributions, including our grammar, the syntax tree, and the active use of generated constraints to refine the generated sketches. We will clarify this when discussing PolyGen in Section 2.
>
> ### Limited primitive and constraint types and simple sketches
> The complexity of the sketches we generate is representative of the complexity available in the SketchGraphs dataset. Like other concurrent papers that use this dataset, we use the official filtered version of the dataset that removes the large number of sketches that are too simple in the full dataset, and also a smaller number of sketches that are too complex. In this filtered dataset, we picked *all* the primitive types and the most common constraint types, corresponding to ~89% of the constraints. We do, however, aim to include more primitive and constraint types in future work.

---

### Official Review · Reviewer_BbqY · 2021-07-26

**Rating:** 5
**Confidence:** 4

**Summary:**

This paper proposes a sequential language to convert the CAD sketches to syntax trees. With
the help of proposed grammar, each sketch is represented as a sequence of primitives (e.g.
lines, arcs) and constraints (e.g. collinear, parallel, tangent). The authors propose a generative
model based on the transformer architecture to generate primitives linked by constraints which
could be optimized by a constraint solver, provided by Onshape. The proposed method is
evaluated on two tasks: a) generating sketches from scratch, and b) auto-constraining
sketches.

**Ethical Concerns:**

No.

**Limitations And Societal Impact:**

While the proposed sequential language for representing the sketches as syntax trees is
helpful for solving the heterogeneous nature between sketch graphs, and the proposed
generative model shows good quantitative results, I think the experiments in this paper are not
sufficient to show the superiority of the proposed method on CAD sketch generation. The
ablation study also does not show a very significant effect of different components of their
methods.

**Main Review:**

*Strengths*:
The paper is well-written and the logic of the paper is smooth. Figure 1 and Figure 3 clearly
state the proposed sequential for representing the sketch and workflow of the generative
model. The sequential language for representing the sketches as syntax trees is a novel way to
solve the heterogeneous nature between different sketch graphs. The result shows the statistical
consistency between the data sketches and the generated sketches via the proposed generative
model. The generative constraints help obviously correcting the quantization error.

*Weakness*:
1. Would it be better to include other generative models for sketches as baselines make this paper
stronger, e.g. Sketch-RNN. In this paper, the baseline is not sufficient.
2. For your ablation study, I think adding the sequence Q3 and/or Q4 does not make a very
significant difference. This makes me question the effectiveness of using this information.
3. I think it is necessary to compare with some non-transformer-based methods to show the
effectiveness of using this architecture in the paper.

**Time Spent Reviewing:**

2 hours

---

> ### Author Response · Authors · 2021-08-10
> **Response to Reviewer BbqY**
>
> ### Additional baselines
> Note that traditional sketch generators like DeepSVG or SketchRNN do not generate heterogeneous primitives with associated *constraints*, thus they cannot accurately model CAD sketches, and do not have the down-stream advantages that CAD sketches provide, like reducing undesired degrees of freedom for easier exploration, easier and more precise constrained editing, etc. Keeping that in mind, we will include a comparison to DeepSVG in the final version. We already made the necessary adjustments to the code of DeepSVG, but did not have enough time to finish running the full experiment in the short time of the rebuttal.
>
> ### Benefits of using the sequences $Q^3$ and $Q^4$
> Removing the sequences $Q^3$ and $Q^4$ does have a significant effect on the results. Especially on sketches with complex primitives and constraints, they help regularize the generation problem by providing some prior knowledge about the structure of the sketch representation. We will add a few qualitative results to illustrate this point.

---

### Author Response · Authors · 2021-08-10
**Response to all reviewers**

We thank all reviewers for their insightful comments and suggestions. We are encouraged to see that reviewers acknowledge that our problem setting of generating sketches with heterogeneous primitives and constraints challenging (reviewers *wERC*, *vzeX*), interesting and rather new (*vzeX*), and our solution with syntax trees, a carefully designed grammar, and actively enforced constraints is novel (*WjPj*, *BbqY*) and, together with concurrent work, a first attempt at solving the problem (*wERC*), showing successful CAD sketch generation (*Wjpj*).

We propose the following changes that we could comfortably fit in the time frame of a minor revision, since we already finished large parts of the required changes/additions to our framework:
- Adding a comparison to DeepSVG.
- Adding an application that identifies primitives and constraints from sketch images, by generating sketches conditioned on sketch images.
- Qualitative results showing the difference between using/not using the sequences $Q^3$ and $Q^4$.
- Add qualitative results of the SketchGraphs baseline.
- Several smaller text clarifications/changes as suggested by the reviewers.

We respond to issues raised by individual reviewers in the comments for each review.

---

### Decision · Program_Chairs · 2021-09-27

**Decision:**

Accept (Poster)

**Comment:**

The paper contributes a generative model for CAD sketches. In particular, the work proposes a language to address the heterogeneity in sketch primitives and constraints, which allows a CAD sketch to be tokenized such that it can be handled using sequence models (such as Transformers) followed by a constraint optimizer. Overall, the reviewers are positive about the paper. The reviewers think the work is well motivated and the paper is clearly written. While several reviewers think the experiments can be strengthened with more comparison, the reviewers generally agree the work is novel and makes reasonable modeling choices, which are valuable to the field. The reviewers raised a set of questions for clarification, which were mostly addressed by the authors in the rebuttal. The authors should further address these points in the revision.